# Particulate Matter Increases the Severity of Bleomycin-Induced Pulmonary Fibrosis through KC-Mediated Neutrophil Chemotaxis

**DOI:** 10.3390/ijms21010227

**Published:** 2019-12-28

**Authors:** I-Yin Cheng, Chen-Chi Liu, Jiun-Han Lin, Tien-Wei Hsu, Jyuan-Wei Hsu, Anna Fen-Yau Li, Wen-Chao Ho, Shih-Chieh Hung, Han-Shui Hsu

**Affiliations:** 1Institute of Emergency and Critical Care Medicine, School of Medicine, National Yang-Ming University, Taipei 112, Taiwan; itoe208@gmail.com (I.-Y.C.);; 2Division of Traumatology, Emergency Department, Taipei Veteran General Hospital, Taipei 112, Taiwan; abby32113@gmail.com; 3Faculty of Medicine, School of Medicine, National Yang-Ming University, Taipei 112, Taiwan; fyli@vghtpe.gov.tw; 4Division of Thoracic Surgery, Department of Surgery, Taipei Veterans General Hospital, Taipei 112, Taiwan; 5Department of Pathology and Laboratory Medicine, Taipei Veterans General Hospital, Taipei 112, Taiwan; 6Department of Public Health, China Medical University, Taichung 404, Taiwan; whocmu@gmail.com; 7Institute of New Drug Development, Biomedical Sciences, China Medical University, Taichung 404, Taiwan; 8Integrative Stem Cell Center, Department of Orthopedics, China Medical University Hospital, Taichung 404, Taiwan

**Keywords:** particulate matter, idiopathic pulmonary fibrosis, neutrophil elastase, sivelestat

## Abstract

Background: Although particular matter (PM) increases incidence and severity of idiopathic pulmonary fibrosis, the underlying mechanism remains elusive. Methods: The effects of PM were evaluated in a murine model of bleomycin-induced pulmonary fibrosis. Mice were divided into four groups, receiving: (1) Saline (control), (2) bleomycin, (3) PM, or (4) bleomycin plus PM (Bleo+PM). Additional groups of Bleo+PM mice were treated with sivelestat (an inhibitor of neutrophil elastase) or reparixin (a C-X-C motif chemokine receptor 2 antagonist), or were genetically modified with keratinocyte chemoattractant (KC) deletion. Results: Pulmonary fibrosis was not observed in the control or PM groups. Bleomycin induced pulmonary fibrosis within 14 days. The Bleo+PM group showed worse pulmonary fibrosis when compared to the bleomycin group. Analyses of immune cell profile and chemokine/cytokine concentrations at day 2-bronchoalveolar lavage fluid (BALF) revealed that the Bleo+PM group had increased neutrophil number and elastase level and KC concentration compared to the bleomycin group. Neutrophil elastase activated the Smad2/Smad3/α-SMA pathway to induce collagen deposition, while sivelestat abrogated the increased severity of pulmonary fibrosis caused by PM. Chemotaxis assay revealed that BALF of the Bleo+PM group recruited neutrophil, which was dependent on KC. Further, genetic KC deletion or pharmaceutical inhibition of KC binding to CXCR2 with reparixin ameliorated the PM-induced increased severity of pulmonary fibrosis. Conclusions: These data provide evidence that the PM-induced increased severity of pulmonary fibrosis depends on KC-mediated neutrophil chemotaxis and give additional mechanic insight that will aid in the development of therapeutic strategies.

## 1. Introduction

Interstitial lung diseases are chronic lung diseases characterized by varying degrees of inflammation and fibrosis. Mostly they are idiopathic, including idiopathic pulmonary fibrosis (IPF), which is a specific disorder characterized by a progressive and irreversible decline in lung function, leading commonly to respiratory failure and fatality [1]. The incidence of IPF is estimated to be 3–9 cases per 100,000 individuals annually in Europe and North America and appears to be increasing over time in most countries [2]. Patients with IPF have a median survival of 2.5–3.5 years after diagnosis due to progressive worsening of lung function [3]. Unfortunately, no clear etiologic factors have been identified, and there are no effective measures for prevention and treatment [4].

Although the exact etiology of IPF remains unknown, recent studies have indicated that a number of environmental factors such as cigarette smoking [5] and wood dusts [6] predispose to the condition. According to United States Environmental Protection Agency, particulate matter (PM) is often considered as one of the main indicators of air pollution and causes serious health effects to the heart and lungs. Especially in China, where the highest annual level of PM has exceeded the primary standard (12 µg/m^3^) 50 folds, air pollution has become a major threat to public health [7]. Airborne PM consists of a heterogeneous mixture of solid and liquid particles, suspended in air, that vary continuously in size and chemical composition in space and time [8]. 

An epidemiological study investigating the association between chronic exposure to NO_2_, ozone (O_3_), and PM with an aerodynamic diameter <10 μm (PM10) and IPF incidence in Northern Italy and detected a significant association between the incidence of IPF and the concentration of NO_2_, but not O_3_ and PM10 [9]. Notably, a study performed in the US proposed that PM10 concentration is associated with an increase in the rate of decline of forced vital capacity (FVC) in IPF [10]. Other studies using Cox proportional hazards model to evaluate the impact of air pollution on acute exacerbation (AE) or disease progression of IPF in longitudinal cohorts identified that onset of AE was significantly associated with antecedent 6-week increases in O_3_ concentration [11]. Besides, mortality was significantly associated with increased levels of exposure to PM2.5 and PM10 [12]. Together, these epidemiological studies argue that air pollution has a negative impact on IPF incidence and outcomes; however, the underlying mechanisms that regulate these processes are still being uncovered.

The current study investigated whether exposure to PM increases the severity of IPF using a well-established murine model of bleomycin-induced pulmonary fibrosis [13,14]. PM was administered via intratracheal instillation to simulate the process of pulmonary fibrosis in patients who live in severe air pollution environments. The effects of PM on pulmonary fibrosis were investigated by lung function test and histological analysis of lung tissues. Immune cell profile and chemokine/cytokine concentrations in bronchoalveolar lavage fluid (BALF) were analyzed to uncover the underlying mechanisms.

## 2. Results

### 2.1. Particulate Matter Increases the Severity of Bleomycin-Induced Pulmonary Fibrosis

Bleomycin and PM were administrated via intratracheal instillation and the subsequent changes in lung function and pulmonary fibrosis 14 days later were assessed using barometric plethysmography and total collagen content in lung tissues, respectively (Figure 1A). Compared to control, mice receiving bleomycin exhibited increased Penh value (Figure 1B) and total collagen content (Figure 1C). Picro Sirius red, and Masson’s trichrome staining further demonstrated increased pulmonary fibrosis (Figure 1D). PM administrated alone did not induce pulmonary fibrosis within 14 days, while PM administrated with bleomycin increased Penh value (Figure 1B) and total collagen content (Figure 1C) when compared to bleomycin alone. Similar data were observed in Picro Sirius red, and Masson’s trichrome staining (Figure 1D). These data suggest that the severity of bleomycin-induced pulmonary fibrosis is increased by exposure to PM.

### 2.2. Particulate Matter Induces Neutrophil Accumulation, Which Releases Neutrophil Elastase to Increase the Severity of Pulmonary Fibrosis

To investigate the underlying mechanism of impact of PM exposure on bleomycin-induced pulmonary fibrosis, the total cell content and immune cell profile in the BALF were first analyzed. Increased polymorphonuclear neutrophils (PMN) count with predominant increase in neutrophils in the Bleo+PM group compared to PM or bleomycin alone groups was noted on day 2 but not on day 7 (Figure 2A,B), suggesting recruitment of neutrophils by PM in mice with bleomycin-induced pulmonary fibrosis in the acute stage with subsidence later on. As neutrophil elastase is required for bleomycin-induced pulmonary fibrosis [15], neutrophil elastase concentrations in BALF were further examined. A significant increase in neutrophil elastase concentration in the Bleo+PM compared to PM or bleomycin alone groups was noted on day 2 but not on day 7 (Figure 2C), which was compatible with the increase in the number of neutrophils.

To examine the involvement of neutrophil elastase in PM-induced increased severity of bleomycin-induced pulmonary fibrosis, we first showed that neutrophil elastase induced differentiation of primary mouse lung fibroblasts into myofibroblasts as demonstrated by the increased expression of α-smooth muscle actin (α-SMA), a marker of myofibroblast [16], after treatment with neutrophil elastase (Figure 2D). Moreover, inhibition of neutrophil elastase with sivelestat (Figure 2E), a neutrophil elastase inhibitor [17], ameliorated PM-induced lung function deterioration (Figure 2F) and pulmonary fibrosis (Figure 2G,H). These data together suggest the involvement of neutrophil recruitment and elastase in increasing the severity of bleomycin-induced pulmonary fibrosis caused by PM.

### 2.3. Neutrophil Elastase Activates the Smad2/Smad3/α-SMA Signaling Pathway 

We further evaluated the molecular pathway that PM or neutrophil elastase activates to increase the severity of bleomycin-induced pulmonary fibrosis. A major process involved in pulmonary fibrosis development is the differentiation of fibroblasts into myofibroblasts [18,19], which requires Smad2/Smad3/α-SMA activation [16]. Western blot analysis of total lung protein revealed upregulated protein levels of phosphorylated Smad2, Smad3, and α-SMA in the Bleo+PM group, compared to control or PM or bleomycin alone group (Figure 3A), which were blocked by treatment with sivelestat (Figure 3A). Moreover, neutrophil elastase activated the Smad2/Smad3/α-SMA pathway in primary lung fibroblasts in a dose-dependent manner (Figure 3B). Histology and IHC study of lung sections further showed that the increased Picro Sirius red stained area was compatible with increased α-SMA stained area in the Bleo+PM group compared to the control group (Figure 3C), which was further blocked by treatment with sivelestat (Figure 3C). These data suggest that PM or neutrophil elastase induce myofibroblast differentiation by activating the Smad2/Smad3/α-SMA pathway.

### 2.4. KC in BALF Recruits Neutrophils

We then investigated the mechanism that PM mediates to recruit neutrophils in bleomycin-induced pulmonary fibrosis. First, a chemotaxis assay was used to evaluate the chemoattractant effects of BALFs collected on day 2 and showed that the Bleo+PM group attracted neutrophils the most (Figure 4A). Furthermore, evaluation of chemokine/cytokine concentrations on day 2-BALF with a protein array revealed KC as the chemokine most enriched in the Bleo+PM group compared to other groups (Figure 4B), which was confirmed by ELISA assays (Figure 4C). Notably, the patterns of KC concentration changes in different groups were similar to that of neutrophil recruitment changes (Figure 4A,C). Interestingly, blocking of KC with corresponding neutralization antibodies significantly suppressed the neutrophil chemoattractant effect of day-2 BALF in the Bleo+PM group (Figure 4D). Since alveolar macrophage is the major source of pro-inflammatory chemokines upon phagocytosis of urban air particles [20], we then examined whether macrophage was activated by PM to secrete KC in bleomycin-induced pulmonary fibrosis. Immunofluorescence revealed the number of cells co-expression of KC and F4/80, a major marker of macrophages [21], on day 2-lung sections was significantly greater in the Bleo+PM group than in the control group (Figure 4E). These data suggest that KC released by macrophage upon PM activation recruits neutrophils in bleomycin-induced pulmonary fibrosis. 

### 2.5. The Increased Severity of Pulmonary Fibrosis Caused by Particulate Matter is Diminished in KC-Deficient Mice

To demonstrate the roles of KC in neutrophil recruitment and consequent the increased severity of pulmonary fibrosis in mice treated with bleomycin and PM, we developed the KC knockout mice (KC^−/−^, KC-deficient). Neutrophil number (Figure 5A) and elastase concentration of day 2-BALF (Figure 5B) were significantly reduced in KC-deficient mice in comparison with wild type littermates. Moreover, KC depletion also significantly improved lung function (Figure 5C) and reduced total collagen content of lungs (Figure 5D) on day 14. H&E, Picro Sirius red, and Masson’s trichrome staining of lung tissue sections also demonstrated less pulmonary fibrosis in KC-deficient mice (Figure 5E). These data together suggest that KC plays a critical role in increasing the severity of bleomycin-induced pulmonary fibrosis caused by PM. 

### 2.6. Inhibition of KC Binding by Reparixin Ameliorates the Increased Severity of Pulmonary Fibrosis Caused by Particulate Matter

To demonstrate whether these findings could be applied for future human application, we first showed that IL-8 (the human functional analogue of KC) [22] and F4/80 were expressed together in areas stained by Picro Sirius red of four individual human pulmonary fibrosis sections (Figure 6A). In mice, KC binds to CXCR2 in neutrophil [23] and this binding is responsible for neutrophil chemotaxis [24]. We then demonstrated that pharmaceutical inhibition of KC binding with reparixin (Figure 6B), a CXCR2 antagonist that has been used in trials for the treatment of breast cancer [25], ameliorated PM-induced increased severity of pulmonary fibrosis. Co-treatment with reparixin in mice receiving PM and bleomycin reduced neutrophil number (Figure 6C) and neutrophil elastase concentration of day 2-BALF (Figure 6D). Moreover, reparixin improved lung function (Figure 6E) and ameliorated pulmonary fibrosis as assayed by total collagen content (Figure 6F) and histochemical stains of fibrosis markers (Figure 6G) on day 14-lung tissues. 

## 3. Discussion

Recent epidemiology studies have highlighted the effects of air pollution on increasing IPF severity. However, the direct roles of air pollution or PM in increasing IPF severity have not been elucidated. The current study demonstrated that PM increases the severity of bleomycin-induced IPF in mice via increasing the number of neutrophils as well as the level of neutrophil elastase in lung tissues. Further, we showed that PM activates macrophages to secrete KC to recruit neutrophils. KC deletion or treatment with reparixin ameliorates PM-induced increased severity of bleomycin-induced pulmonary fibrosis. More importantly, we showed that IL-8 and F4/80 were expressed in the same areas of human lung tissues with pulmonary fibrosis. 

Bleomycin-induced pulmonary fibrosis in mice is one of the most used animal models of IPF [13]. Based on the classification in previous experiments, the process of pulmonary fibrosis is divided into two stages. The early stage is the inflammatory phase, at around 0–7 days after bleomycin administration, while the later stage is the fibrotic phase, starting from 7 days of bleomycin administration [26]. According to these information, we checked the severity of pulmonary fibrosis at day 14 but explored the effects of PM on inflammatory phases by specifically evaluating the changes in immune cell number and profile in BALF between days 2 and 7. In the BALF of the Bleo+PM group, large accumulation of neutrophils and abundant levels of neutrophil elastase were observed on day 2 of co-administration of bleomycin and PM. We further demonstrated in vitro that neutrophil elastase activated the Smad2/Smad3/α-SMA pathway to induce differentiation of primary mouse lung fibroblasts into myofibroblasts. More importantly, treatment with sivelestat almost completely blocked this pathway and ameliorated PM-mediated the increased severity of bleomycin-induced pulmonary fibrosis. Similarly, clinical data indicated an increase in the total number of neutrophils in BALF of IPF patients with onset of AE [27]. Additionally, incremental increase in neutrophil elastase was detected in patients with IPF [28]. These data suggest the essential roles of neutrophil accumulation and increased neutrophil elastase level in BALF in the development and progression of IPF. It should be noted that when PM was administrated 7 days (during the fibrotic phase) after bleomycin instillation, it did not worsen the pulmonary fibrosis induced by bleomycin (Appendix A). Thus, PM only increases the severity of pulmonary fibrosis in the early stage (the inflammatory phase) but not in the late stage (the fibrotic phase) suggesting that PM increasing the severity of pulmonary fibrosis is time- or phase-dependent. 

After bleomycin instillation, KC was detected as early as 6 h later and returned to basal levels by around 1 week later [29]. The relevant roles of KC and its receptor, CXCR2, in the development of pulmonary fibrosis induced by bleomycin have also been demonstrated [29]. The involvement of KC in the increased severity of bleomycin-induced pulmonary fibrosis caused by PM has also been demonstrated by the current study, where KC was the chemokine most elevated in day 2-BALF of mice receiving bleomycin and PM when compared with that of mice receiving bleomycin alone. Moreover, KC depletion or treatment with reparixin blocked PM-induced neutrophil accumulation and consequent worsening of lung function and pulmonary fibrosis. We further showed that KC or its human analogue IL-8 was expressed by cells positive for macrophage marker F4/80 in murine or human pulmonary fibrosis tissues. These data suggest the synthesis and secretion of KC and IL-8 by murine or human macrophages, which is supported by a study that showed KC is newly synthesized by tissue macrophages when stimulated with LPS [30].

Although we demonstrated and elucidated the mechanism that by which PM mediates increased severity of bleomycin-induced pulmonary fibrosis; however, there are still limitations to the current study, including study designs. One of the limitations is the PM particles, urban dust 1649b, used in this study, which was collected in the 1970s. It is anticipated that PM collected in 1970s has a different composition to that in the present day, although urban dust 1649b is one of the most analyzed and certificated PM particles [31]. The other is the high dose of PM used in the study. Dose-response is an important component in risk assessment due to the nature of animal toxicological studies, with high dose having high response, and further to extrapolate and assess potential human exposure and risk. Especially in air toxics, where epidemiologic and toxicological data have typically resulted from exposure levels that were high relative to environmental levels. For investigation of the influence of PM on murine pulmonary fibrosis in the short term, we therefore determined a high dose that was considered acceptable in this study. 

## 4. Materials and Methods

### 4.1. Reagents

PM (NIST^®^ SRM^®^ 1649b, Sigma-Aldrich, St. Louis, MO, USA) was prepared from atmospheric particulate material collected in the Washington, DC area in 1976 and 1977 using a baghouse specially designed for the purpose. This particulate matter (PM) was collected over a period in 12 months, and therefore represents a time-integrated sample. The detailed information of the PM is available on the manufacture’s website. In brief, PM was resolved in sterile phosphate-buffered saline (PBS) and sonicated before instillation to ensure an even distribution and prevent aggregation of the particles. Bleomycin (Sigma-Aldrich, St. Louis, MI, USA) was resolved in PBS. The distribution of the particles was checked in the lung of mice after 14 days of receiving PM (Appendix A).

### 4.2. Animal Models

All animal experiments were conducted in accordance with the committee guidelines of the Institutional Animal Care and Use Committee of Taipei Veterans General Hospital (Protocol IACUC number: 2017-003, 1 January 2017). Eight-week-old male C57BL/6 mice were purchased from BioLASCO Taiwan Co., Ltd. (Taipei, Taiwan). Mice were divided into four groups, receiving (1) vehicle control; (2) PM (200 µg/mouse); (3) bleomycin (2 U/kg); and (4) bleomycin plus PM, respectively. After instillation of each aliquot of vehicle or PM or/and bleomycin, the mice were placed in right and then left lateral decubitus position for 10–15 s to facilitate equal distribution of the reagents. Pulmonary fibrosis was induced by intratracheal administration of 2 U/kg body weight of bleomycin (Sigma-Aldrich, St. Louis, MI, USA) in 50 µL of sterile PBS. The control group received the same volume of sterile PBS. Animals were sacrificed on days 2, 7, and 14 for further analyses. Additional groups of mice receiving bleomycin plus PM were used for testing treatment strategies. After administration of bleomycin plus PM, mice were divided into two groups, receiving vehicle and sivelestat, respectively. Sivelestat treatment protocol was followed as described previously [17]. The treatment group received intraperitoneal injection of sivelestat loading dose (100 mg/kg in 200 µL sterilized saline) (Tocris, Bristol, U.K.) on day 1, followed by maintenance dose (10 mg/kg in 200 µL sterilized saline) from days 2 to 7. The control group received DMSO in 200 µl sterilized saline from day 1 to day 7. In another treatment test, mice were divided into two groups, receiving vehicle and reparixin, respectively after administration of bleomycin plus PM. Reparixin treatment protocol referred to that described in previous research [32,33,34]. The treatment group received subcutaneous injection of reparixin (30 mg/kg in 100 µL saline-diluted DMSO) (Tocris, Bristol, U.K.) at 30 min before administration of bleomycin and PM, followed by maintenance dose (30 mg/kg in 100 µL saline-diluted DMSO) from day 0 to day 2. The control group received saline-diluted DMSO from day 1 to day 2. 

### 4.3. Generation of Keratinocyte Chemoattractant (KC/CXCL1) Knockout Mice 

CRISPR/Cas9 technology was used to generate KC-deficient mice. Paired sgRNA oligos were designed, one for guiding Cas9 cleavage in the upstream sequence of exon 1 and the other, downstream of the last exon of KC. The sgRNA Designer: CRISPRko [35] and the Cas-OFFinder [36] were applied to find the sgRNA sequences with few or no related sites in the genome. The upstream and downstream sgRNA target sequences with PAM sites (NGG) were 5′-GAGATGCTGCGGATACAGGGAGG-3′and 5′-ACACTGTGAAGTAAAATACGTGG-3′, respectively. The length of the deleted region was about 5.4 kb. The sgRNA and Cas9 RNA for microinjection were prepared following the commercial protocol, for the AmpliCap-MaxTM T7 High Yield Message Maker kit (C-ACM04037, CELLSCRIPT, Madison, WI, USA). Pronuclear microinjection was performed on fertilized eggs from C57BL/6J mice. KC genotyping was performed by PCR (Appendix A). The primer sequences for identifying the WT and deleted alleles, respectively were 5′-GGTCCATTAATGTCCACTAC-3′ paring with 5′-TGGATGGC AGTGATGGTTGC-3’ and 5′-GGTCCATTAATGTCCACTAC -3′ paring with 5′-TCCTGGAACTCACTCTGTAG-3′. All techniques for production of the KC-deficient mice were provided by the Transgenic Mouse Model Core Facility (supported under grants from National Core Facility for Biopharmaceuticals, Ministry of Science and Technology, Taiwan).

### 4.4. Measurement of Lung Function

Lung function was measured in unrestrained mice using barometric whole-body plethysmography (Buxco1; EMKA Technologies, Paris, France) before drug injection and at 14 days after administration of PM or/and bleomycin. The Penh value was read by Biopac AcqKnowledg software (BIOPAC, Goleta, CA, USA).

### 4.5. Collagen Content Assay

Total lung collagen content was measured using Total Collagen Colorimetric Assay Kit (BioVision, Milpitas, CA, USA) according to the manufacturer’s protocol. In brief, every 10 mg of lung tissue was homogenized in 100 µL ddH_2_O and mixed with equal volume 12M HCL. Then, samples were hydrolyzed at 120 °C for 3 h. After homogenization, 4 mg of activated charcoal was added into individual tubes and vortexed. After centrifuging at 10,000× *g* for 3 min, 15 µL supernatant of each vial was transferred to a 96-well plate and dried in a 70 °C oven. Reaction reagent was added into each well and absorbance was measured at 560 nm using a SpectraMax M5 multi-mode reader (Molecular Devices, San Jose, CA, USA)

### 4.6. Preparation and Analysis of Bronchoalveolar Lavage Fluid (BALF)

BALF was obtained by intratracheal injection of 1 mL PBS, followed by withdrawal thrice, with one further repetition of the whole process. Totally, 2 mL PBS was injected and about 1.7 mL BALF was recovered. The BALF was centrifuged at 300× *g* for 5 min at 4 °C, the supernatant was collected for ELISA assay, chemokine protein array analysis and chemotaxis assay. The pellet was resuspended in 600 µL PBS, and 200 µL aliquots were spun onto a slide with a cytospin centrifuge. The slide was stained with Liu’s stain for counting and classifying the immune cells. The total PMN granulocytes in BALF were stained by trypan blue, and then counted with hemocytometer.

### 4.7. Chemokine Protein Array Analysis

Relative expression levels of chemokines/cytokines in BALF were analyzed by C-Series Mouse Inflammation Antibody Array 1 Kit (RayBiotech, Peachtree Corners, GA, USA) according to the manufacturer’s protocol. After blocking with blocking buffer, 1 mL BALF was pipetted into each well and placed at 4 °C overnight (Four groups). On day 2, membranes were incubated in Biotinylated Antibody Cocktail and HRP-Streptavidin. Finally, the signals were detected with detection buffer and CCD camera. 

### 4.8. Enzyme-Linked Immunosorbent Assay (ELISA)

The concentration of KC and neutrophil elastase (Biorbyt, Cambridge, U.K.) were determined by ELISA kits according to manufacturers’ protocols.

### 4.9. Hematoxylin-Eosin (H&E), Masson’s Trichrome, and Picro Sirius Red Staining

Mice lungs were fixed by 4% paraformaldehyde perfusion through intratracheal instillation and soaked in 4% paraformaldehyde overnight. Lungs were embedded in paraffin, and the 4-μm-thick tissue sections were stained with H&E, Picro Sirius red (Sigma-Aldrich, St. Louis, MI, USA), and Masson’s trichrome (Sigma-Aldrich). The area of collagen was the red color in the Picro Sirius red stain image. For Masson’s trichrome stained sections, the area of fibrosis was defined as the area of blue. The sections were quantified using a light microscope attached to an image-analysis system (Image-Pro Plus; Media Cybernetics, Rockville, MD, USA).

### 4.10. Cell Cultures

Primary lung fibroblasts were isolated from 4-week-old C57BL6/J mice lung as previously described [37]. The cells were grown in DMEM medium (Gibco, Waltham, MA, USA) supplemented with 10% fetal bovine serum (FBS) containing 100 U/mL penicillin G and 100 µg/mL streptomycin at 37 °C in an incubator containing 5% CO_2_. MPRO Cell Line was obtained from the ATCC (Manassas, Virginia, USA) and grown in Iscove’s modified Dulbecco’s medium (IMDM, Gibco, Waltham, MA, USA) with 4 Mm l-glutamine adjusted to contain 1.5 g/L sodium bicarbonate containing 10 ng/mL murine granulocyte macrophage colony stimulating factor, 80%, and heat-inactivated horse serum, 20% (Gibco, Waltham, MA,, USA). MPRO cells were differentiated into neutrophils from a starting density of 5 × 10^5^ cells/mL with a final concentration of 10 µM all trans-RETINOIC ACID (ATRA; Sigma-Aldrich, St. Louis, MI, USA) in IMDM for 3 days.

### 4.11. Western Blot Analysis

Aliquots of cell lysates (40 µg/lane) from each sample were loaded on 10% SDS-PAGE, electrophoresed, and transferred onto PVDF membrane. Non-specific binding of antibodies was prevented by incubating membranes in 5% milk in TBST for 1 h at room temperature. The membranes were then incubated overnight at 4 °C with first antibodies against pSmad3 (Ser423/425) (1:1000 dilution), Smad3 (1:1000 dilution), pSmad2 (Ser465/467) (1:1000 dilution), Smad2 (1:1000 dilution), GAPDH (1:20,000 dilution) (Cell Signaling Technology, Danvers, MA, USA), and α-SMA (1:3000 dilution) (Abcam, Cambridge, U.K.). After washing three times with TBST, the membranes were incubated for 1 h at room temperature with HRP-labeled secondary antibody (anti-rabbit, or anti-goat; 1:10,000 dilution). After thorough washing with TBST, the signals were detected with a chemiluminescent system (PerkinElmer Life and Analytical Sciences, Boston, Massachusetts, USA). Membranes were exposed to X-ray film to visualize the bands (Amersham Pharmacia Biotech, Piscataway, NJ, USA). 

### 4.12. Immunohistochemistry (IHC)

The slides were deparaffinized and rehydrated with Xylene and different concentrations of ethanol. Sections were permeabilized with 0.1% Triton X-100 in PBS. After quenching with 3% hydrogen peroxide for 20 min, cells were blocked with 1% milk in PBS, and then incubated overnight with first antibody, anti-α-SMA (1:50 dilution), anti-F4/80 (1:100 dilution), or anti-IL-8 (1:200 dilution) at 4 °C. After incubation, cells were washed with PBS twice and tagged with secondary antibody (HRP anti-goat, 1:1000 dilution) for 1 h at room temperature.

### 4.13. Immunofluorescence

Sections from day 2 scarified mouse lung were deparaffinized and rehydrated with Xylene and different concentration ethanol. The slides were permeabilized with 0.1% Triton X-100 in PBS. After washing slides with PBS three times, sections were blocked with 5% FBS in 0.1% PBS, and then incubated overnight with first antibody, anti-F4/80 (Abcam) and anti-KC (R&D Systems, Bio-Techne, Minneapolis, MI, USA) (1:100 dilution) at 4 °C. After incubation, slides were washed with PBS twice and tagged with secondary antibody Alexa Fluor 488 (green) or Alexa Fluor 647 (red) (Abcam)for 1 h at room temperature. Slides were mounted with DAPI and imaged by fluorescence microscopy.

### 4.14. Chemotaxis Assay

MPRO cells (murine promyelocytes; available from ATCC clone 2.1 no. CRl-11422) were seeded at a starting density of 3 × 10^5^ cells/mL and differentiated for 3 days into neutrophils with a final concentration of 10 μM all-trans retinoic acid (Sigma-Aldrich). Aliquots of 2 × 10^5^ neutrophils (ATCC, Manassas, VI, USA) in 200 µL HBSS (1% BSA) (Sigma-Aldrich, St. Louis, MI, USA) were cultured in the upper wells of a 24-well Transwell plate (Corning Inc., Corning, NY, USA), while aliquots of 600 µL HBSS (25% day 2-BALF + 1% BSA) were placed in the lower chambers. All cells were incubated in 37 °C, 5% CO_2_ for 6 h. Neutrophils would move through the 3-µm-pore membrane toward chemoattractant underneath the membrane. Lower chamber suspension was centrifuged at 100× *g* for 7 min. After the removal of supernatant, the cells were then resuspended in 100 µL PBS. We added 10 µL trypan blue to the 10 µL cell suspension, and then counted the number of neutrophils with hemocytometer. For the antibody neutralization experiments, anti-KC or isotype IgG (R&D Systems) were mixed with 600 µL HBSS (25% day2 BALF + 1% BSA) overnight at 4 °C and then added into the lower chamber for further chemotaxis assay.

### 4.15. Immunocytochemistry

Fibroblasts treated without or with 8 nM neutrophil elastase (R&D SYSTEMS, Minneapolis, MI, USA) were fixed with 4% paraformaldehyde (Sigma-Aldrich, St. Louis, MO, USA) in PBS and then washed with PBS twice. Cells were permeabilized with 0.2% Triton X-100 (Sigma-Aldrich, St. Louis, MO, USA) in PBS. After washing cells with PBS twice, cells were blocked with 1% FBS in 0.1% PBS, and then incubated overnight with first antibody, anti-α-SMA (1:100 dilution) (Abcam), at 4 °C. After incubation, cells were washed with PBS twice and tagged with secondary antibody (Alexa Fluor 488, anti-rabbit, 1:200) for 1 h at room temperature. Slides were mounted with DAPI (Invitrogen, Carlsbad, CA, USA) and imaged by fluorescence microscopy.

### 4.16. Human Lung Tissue Sections

Four pulmonary fibrotic tissue blocks were from patients who had been diagnosed as usual interstitial pneumonia and received surgical resection of lung in 2000–2018. All experiments were approved by the Institutional Review Board of Taipei Veterans General Hospital (Protocol IRB number: 2019-12-005CC, 15 August 2019). The demographic data of blocks and the pictures with smaller magnifications (40×) are shown in Appendix A.

### 4.17. Statistical Analysis

Values were reported as the means ± standard deviations. Student’s *t*-test and one-way ANOVA with Dunnett’s test were employed for two-group and >3-group comparisons, respectively. *p* < 0.05 was considered statistically significant.

## 5. Conclusions

We demonstrate that the severity of bleomycin-induced pulmonary fibrosis is increased by PM, which activates macrophages to secrete KC, thereby increasing neutrophil number and neutrophil elastase level. Consequently, neutrophil elastase activates the Smad2/Smad3/α-SMA pathway to induce myofibroblast differentiation and pulmonary fibrosis. More importantly, we demonstrate that treatment with sivelestat and reparixin can block PM-induced pulmonary fibrosis. 

## Figures and Tables

**Figure 1 ijms-21-00227-f001:**
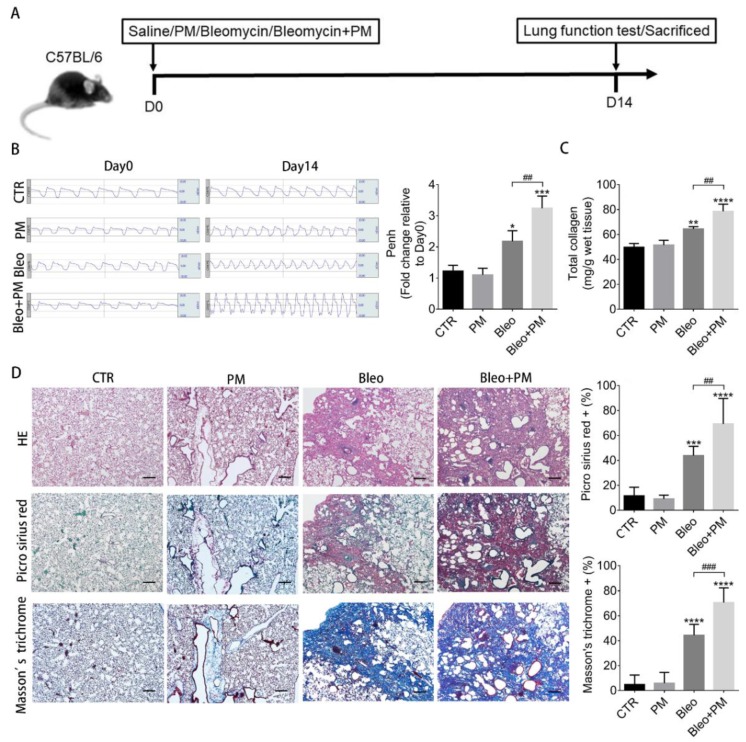
Particulate matter (PM) enhances lung function deterioration and pulmonary fibrosis. (**A**) C57BL/6 mice were intratracheally instilled with saline (Control), 200 µg PM, 2 U/kg bleomycin (Bleo) or 200 µg PM plus 2 U/kg bleomycin (Bleo+PM) at day 0. (**B**) The lung function test for the Penh value was performed at 14 days. The mice were then sacrificed, and the lung tissues were subjected to (**C**) total collagen content measurement and (**D**) histochemical analysis with H&E, Picro Sirius red, and Masson’s trichrome staining (40×). (**B**,**C**) Quantification data are expressed as the mean ± SD of three mice in each group. (**D**) Quantification data are expressed as the mean ± SD of six mice in each group. (scale bar: 200 µm) * *p* < 0.05, ** *p* < 0.01, *** *p* < 0.005, **** *p* < 0.001 versus CTR as determined by one-way ANOVA. ## *p* < 0.01, ### *p* < 0.05 as determined by one-way ANOVA.

**Figure 2 ijms-21-00227-f002:**
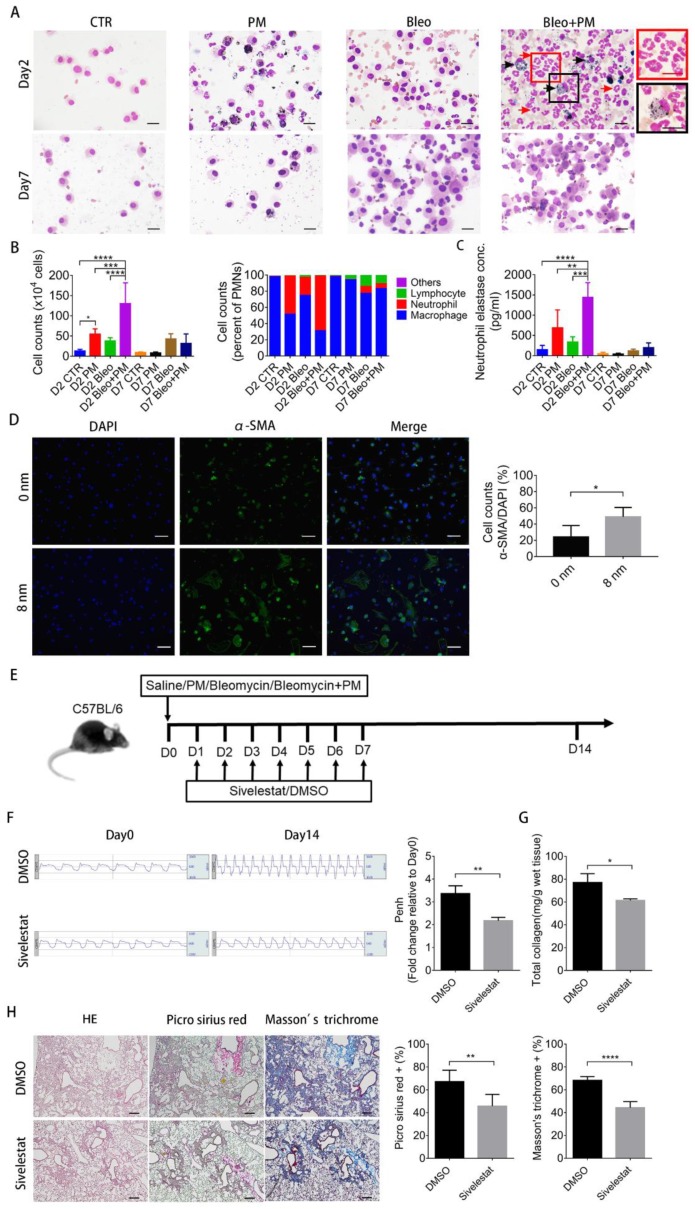
Neutrophil elastase is involved in PM-enhanced lung function deterioration and pulmonary fibrosis. (**A**–**D**) C57BL/6 mice were intratracheally instilled with saline (Control), 200 µg particulate matter (PM), 2 U/kg bleomycin (Bleo) or 200 µg PM plus 2 U/kg bleomycin (Bleo+PM) at day 0. The mice were sacrificed and the 2-bronchoalveolar lavage fluid (BALF) were harvested at 2 and 7 days for (**A**) Liu’s staining (40×) and (**B**) quantification of the changes in the immune cell profiles. Red arrows, frame indicates neutrophil and Black arrow, frame indicates macrophage engulfing PM (scale bar: 20 µm). (**C**) The concentration of neutrophil elastase was measured by ELISA. (**D**) Immunocytochemistry of α-smooth muscle actin (α-SMA) protein in murine lung fibroblast without or with treatment with neutrophil elastase at 8 nm for 1 h (400×) (scale bar: 20 µm) (**E**) Additional groups of C57BL/6 mice were intratracheally instilled with 200 µg PM plus 2 U/kg bleomycin (Bleo+PM), followed by treatment without or with sivelestat (100 mg/kg on day1, 10 mg/kg on day 2–7). (**F**) The lung function test of mice treated with or without sivelestat was performed at 14 days. The mice were then sacrificed and the lung tissues were subjected to (**G**) total collagen content measurement and (**H**) histochemical analysis with H&E, Picro Sirius red, and Masson’s trichrome staining (40×) (scale bar: 200 µm) (**B**) Quantification data are expressed as the mean ± SD of seven mice in each group. (**C**) Quantification data are expressed as the mean ± SD of five mice in each group. (**D**) Quantification data are expressed as the mean ± SD of four independent slides. (**F**,**G**) Quantification data are expressed as the mean ± SD of three mice in each group. (**H**) Quantification data are expressed as the mean ± SD of five mice in each group. (**B**,**C**) ** *p* < 0.01, *** *p* < 0.005, **** *p* < 0.001 as determined by One-Way ANOVA with Tukey’s multiple comparisons test. (**F**–**H**) * *p* < 0.05, ** *p* < 0.01, **** *p* < 0.001 as determined by Student’s *t* test.

**Figure 3 ijms-21-00227-f003:**
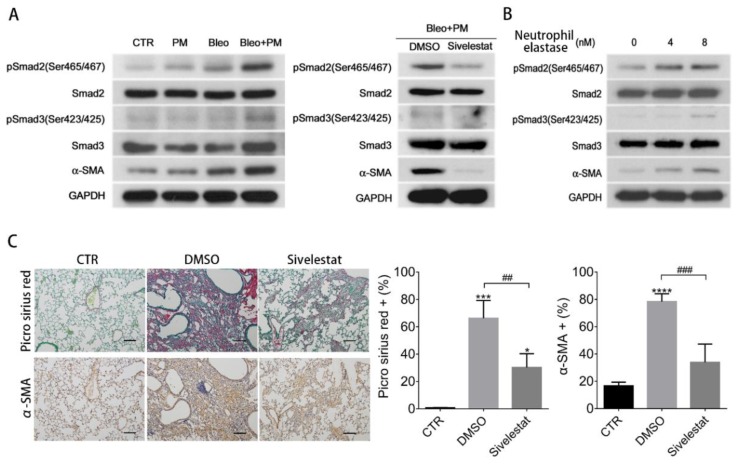
PM-enhanced pulmonary fibrosis caused by neutrophil elastase is through Smad2/Smad3/α-SMA activation. (**A**) Mice were intratracheally instilled with saline (Control), 200 µg particulate matter (PM), 2 U/kg bleomycin (Bleo), or 200 µg PM plus 2 U/kg bleomycin (Bleo+PM). Mice in Bleo+PM group were also treated without (vehicle: DMSO) or with sivelestat (100 mg/kg on day 1, 10 mg/kg on day 2–7). The mice were then sacrificed on day 14 and the lung tissues were subjected to Western blot. (**B**) Primary murine lung fibroblasts were treated with neutrophil elastase at indicated concentration for 1 h, followed by western blot analysis. (**C**) The lung sections of mice with indicated treatment were used for histochemical analysis with Picro Sirius red and immunohistochemistry for α-SMA expression (100×) (scale bar: 100 µm) Quantification data are expressed as the mean ± SD of three mice in each group. * *p* < 0.05, *** *p* < 0.005, **** *p* < 0.001 as determined by one-way ANOVA with Tukey’s multiple comparisons test versus control. ## *p* < 0.01, ### *p* < 0.05 as determined by one-way ANOVA.

**Figure 4 ijms-21-00227-f004:**
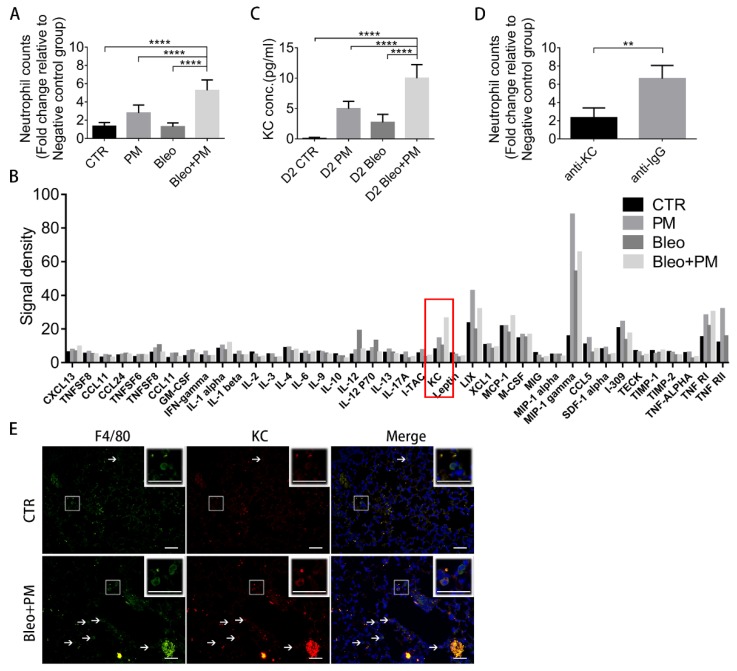
Keratinocyte chemoattractant (KC) depletion diminishes PM-enhanced lung function deterioration and pulmonary fibrosis. Mice were intratracheally instilled with saline (Control), 200 µg particulate matter (PM), 2 U/kg bleomycin (Bleo) or 200 µg articulate matter plus 2 U/kg bleomycin (Bleo+PM). The mice were sacrificed and the BALF were harvested at 2 days for (**A**) chemotaxis assay of neutrophils, (**B**) chemokine protein array (Red frame indicates the signal density of KC) and (**C**) ELISA assay of KC. (**D**) The BALF harvested from the Bleo+PM group at 2 days were pre-treated with anti-KC antibodies or with isotype IgG overnight at 4 °C, followed by chemotaxis assay of neutrophils. (**E**) Mice of the Bleo+PM group were sacrificed at 2 days and then the lung tissue sections were analyzed with double immunofluorescence of F4/80 and KC (200×) white frame areas are magnified on the right upper corners. White arrows indicate macrophages that express F4/80 and KC. (A, C) Quantification data are expressed as the mean ± SD of BALF samples from six mice in each group. (D) Quantification data are expressed as the mean ± SD of four BALF samples from six mice in each group. (scale bar: 50 µm) (**A**,**C**), **** *p* < 0.001 as determined by one-way ANOVA with Tukey’s multiple comparisons test. (**D**) ** *p* < 0.01 as determined by Student’s *t* test.

**Figure 5 ijms-21-00227-f005:**
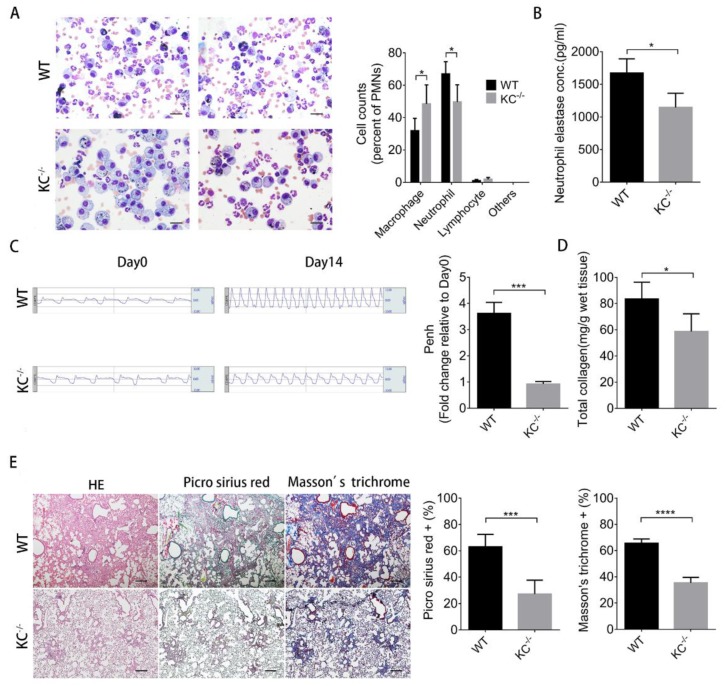
Increased severity of bleomycin-induced pulmonary fibrosis caused by particulate matter (PM) was diminished in KC knockout mice. KC knockout mice (KC^−/−^) and wild-type littermates (WT) were intratracheally instilled with 200 µg particulate matter plus 2 U/kg bleomycin. (**A**,**B**) The mice were sacrificed and the BALF were harvested at 2 days for (**A**) Liu’s staining (400×) and quantification of the changes in the immune cell profile. (scale bar: 20 µm) (**B**) The concentration of neutrophil elastase was measured by ELISA. (**C**) The lung function test for Peng value was performed at 14 days. The mice were then sacrificed and the lung tissues were subjected to (**D**) total collagen content measurement and (**E**) histochemical analysis with H&E, Picro Sirius red, and Masson’s trichrome staining (40×) Quantification data are expressed as the mean ± SD of four mice in each group. (scale bar: 200 µm) * *p* < 0.05, *** *p* < 0.005, **** *p* < 0.001 as determined by Student’s *t* test.

**Figure 6 ijms-21-00227-f006:**
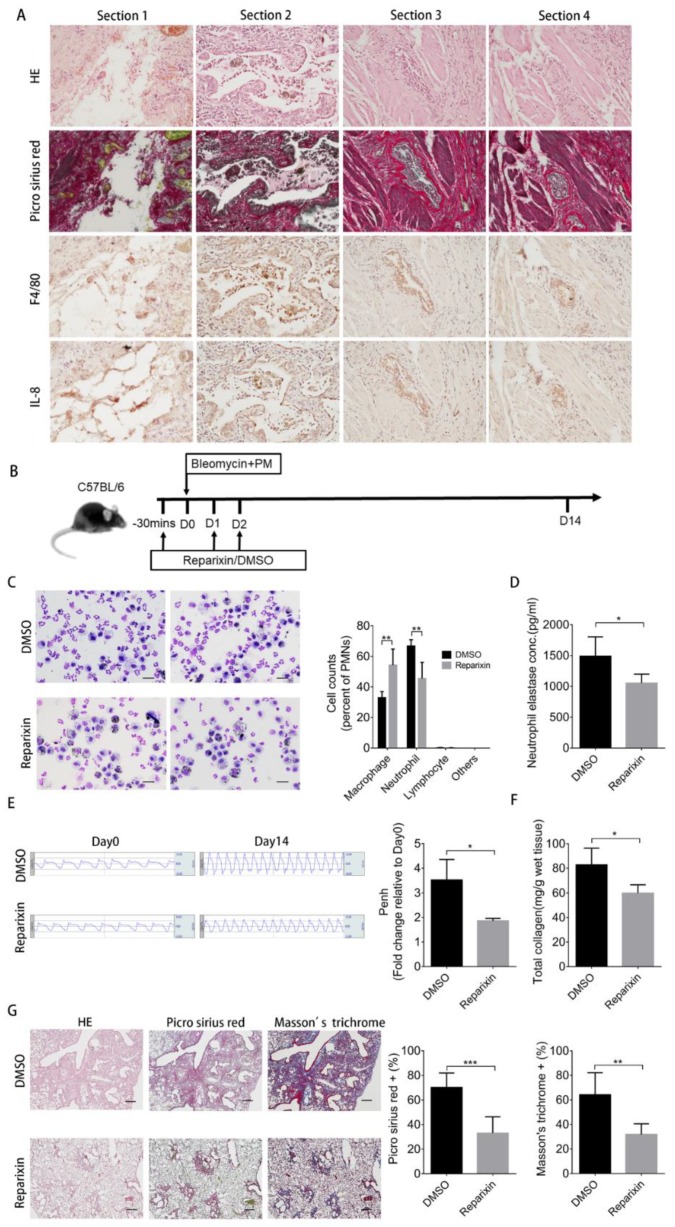
Reparixin ameliorates PM-enhanced pulmonary fibrosis and lung function deterioration. (**A**) Human lung tissue sections from four individuals with pulmonary fibrosis were subjected to immunohistochemistry for F4/80 and IL-8 expression (200×) (scale bar: 50 µm) (**B**) Reparixin-treated mice and DMSO-treated mice were intratracheally instilled with 200 µg particulate matter plus 2 U/kg bleomycin. The mice were sacrificed and the BALF were harvested at 2 days for (**C**) Liu’s staining (400×) and quantification of the changes in the immune cell profile. (scale bar: 20 µm) (**D**) The concentration of neutrophil elastase was measured by ELISA. (**E**) The lung function test for the Penh value was performed at 14 days. The mice were then sacrificed and the lung tissues were subjected to (**F**) total collagen content measurement and (**G**) histochemical analysis with H&E, Picro Sirius red, and Masson’s trichrome staining (40×). Quantification data are expressed as the mean ± SD of four mice in each group. (scale bar: 200 µm) * *p* < 0.05, ** *p* < 0.01, *** *p* < 0.005 as determined by Student’s *t* test.

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
