# Peer review of "Particulate Matter Increases the Severity of Bleomycin-Induced Pulmonary Fibrosis through KC-Mediated Neutrophil Chemotaxis"

_ijms, 2019, doi:10.3390/ijms21010227_

Round 1

Reviewer 1 Report

Congratulation for  your hard work!

An interesting and valuable paper.

The article related wit  Particulate matter exacerbates bleomycin-induced pulmonary fibrosis through KC-mediated neutrophil chemotaxis is:

a very strong article with a very good introduction a strong statistical data with valuable and new references very good images  an interesting approach of the subjects with a valuable conclusion section

Author Response

Response to Reviewer 1 Comments

Point 1: Congratulation for your hard work!

An interesting and valuable paper.

The article related wit Particulate matter exacerbates bleomycin-induced pulmonary fibrosis through KC-mediated neutrophil chemotaxis is:

a very strong article with a very good introduction a strong statistical data with valuable and new references very good images an interesting approach of the subjects with a valuable conclusion section

Response 1: Thank you very much for your kind comments. We are glad to know that you are interested in our research. Moreover, the manuscript has English editing again.

Reviewer 2 Report

This is an interesting and very extensive, thorough body of work exploring the potential of air pollutants to contribute to the pathogenesis of pulmonary fibrosis. The strengths of the paper are in the range of evidence used to identify potential mechanisms. The KC knock out mice, and the sivelestat and reparixin experiments, add significant weight to the findings. The pSMAD experiments are persuasive. The ambition to incorporate lung function data is also a strength (though there is an important caveat, discussed below). The paper therefore provides some fairly robust evidence to implicate KC and neutrophil elastase as determinants of pulmonary fibrosis induced by the simultaneous administration of very high concentrations of PM10 and bleomycin, in bolus form, to the lung, in mice. 

Perhaps inevitably, in such a comprehensive paper using mice to model a human condition, there are a number of potential limitations. 

MAJOR

1. The clinical extrapolation inferred by the authors is extremely hard to justify, and the authors very significantly over-interpret the clinical relevance of their findings, in my opinion.

2. It could be argued that the authors' most interesting data are actually in the supplement, where it appears that PM10 administered AFTER the onset of bleomycin-induced injury, actually REDUCES pulmonary fibrosis. The authors seem to play this down. However, it is almost clinically inconceivable that clinicians could identify patients at the time of their first (pre-clinical) exposure to a pro-fibrotic stimulus (but this is the scenario in their experiments, i.e. bleomycin and PM10 are given together, and in this setting PM10 seem to make fibrosis worse). Much more clinically relevant is the patient in whom fibrosis is already underway in the lungs, and who is then exposed to pollution (the situation in the supplement, when PM10 seem to be associated with LESS fibrosis). 

3. Following on from the above, the human IPF biopsy samples are unconvincing. The reason the samples were taken is not clear, but "resection" implies these were explants removed at the time of lung transplantation? The sections shown are very small, and are not of sufficient size or characteristics to make a diagnosis of UIP, to my mind. 

4. One significant problem is that, as I understand the paper, the authors' hypothesis is that air pollution increases exacerbations of IPF? At no stage do their models show the features of acute exacerbations of IPF. 

5. Their statement in the first paragraph of the discussion, that the bleomycin model is "generally recognized as a standard animal model of IPF" is definitely not true, in my opinion. There are many good reviews that highlight the differences between IPF and the bleomycin model. Even if their statement was true, the bleomycin model does not show the histological and pathological changes most associated with acute exacerbations of IPF. 

6. The authors briefly discuss whether the composition of urban dust from the 1970s can be compared to modern air pollution. What they do not discuss is (a) whether this huge single, bolus inoculum is in any way representative of the sustained low dose pollution that patients are exposed to? (b) why they give an intratracheal instillation, rather than a more sustained nebulised exposure?)?  (c) whether they performed any chemical checks to ensure that the PM10 really was PM10, (d) did they check the distribution of the particles in the murine lung?, and (e) why they did not use contemporary urban dust? 

7. The involvement of KC is quite persuasive. However, it seems very strange that the authors did not include other major murine neutrophil chemoattractants in their array, such as MIP-2 and CXCL1. It seems highly likely that these would have been much more informative than some of the proteins covered in their array. How the array was designed/chosen is unclear.

8. The attempt to incorporate lung function is extremely commendable. However, the authors state in the methods that the enhanced pause method was used to assess "in vivo airway obstruction". Pulmonary fibrosis is characterized by a restrictive ventilatory defect, and airways obstruction is not a characteristic feature. This again raises very serious concerns about the relevance of their model. 

9. The suggestion that their data justify clinical trials of drugs like reparixin in patients with IPF is wildly over-stated, in my view.

MINOR

1. The figures describe "x independent experiments". Does this mean that there were x animals in each group? Or does it mean that the whole experiment was carried out more than once, with y animals studied each time? Either way, the number of animals studies is not clear. It is of some concern that the numbers described differ in each experiment. For example in figure 1 the lung function has "3 independent experiments" the pathology has 6, but I thought that mice were killed after the lung function, so the numbers should be the same?

2. In figure 2, the lung function data are from "4 independent experiments" the histology from 5. The justification of sample size is never stated, and the actual n in each group is not clear. 

3. It is not clear why experiments were concluded at day 7 in figure 2 and at day 14 in figures 1 and 3. In general the timings of the 2 well-described phases of the bleomycin model (a neutrophilic phase and a fibrotic phase) are not well described or considered in relation to their experiments. 

4. The authors describe mice being "randomly" assigned to groups. What do they mean by this? Presumably these were not formally "randomised"? If they were, this should be stated.

5. Some of the neutrophil experiments could be questioned. In the chemotaxis assays, 6 hours is a long time to run the assay, given the tendency for apoptosis to occur in a good proportion of cells in that time scale. It is customary to have a positive and negative control in such experiments. It is a pity that the authors studied neutrophil elastase antigen and not activity (which would have been more informative). I was not sure how or why the authors used trypan blue to identify neutrophils - it is more often used as an indicator of cell viability, not to identify neutrophils. 

6. Some of the arrows used in figures are very hard to make out. In general the micrographs are very small.

Author Response

Thank you very much for your kind comments.

Please see the attachment and refer to our point-by-point responses to the your comments.

Reviewer 3 Report

Comments to authors

The authors have undertaken a study evaluating the effect of PM on bleomycin induced pulmonary fibrosis in a murine model. Their data suggest that the combination of PM and bleomycin, when administered together but not at separate intervals, increases BAL neutrophils and neutrophil elastase and is associated with an increase in pulmonary fibrosis. These effects can be reversed by inhibition of neutrophil elastase by sivelestat or by deletion or blockade of KC by reparixin.

Whilst this work is novel, I have some concerns about the model used as it is not representative of an acute exacerbation of IPF. In its current format, the model of co-administration of PM and bleomycin results in an increase in pulmonary fibrosis but there are no features presented to support an acute exacerbation of pulmonary fibrosis. Of particular importance is the lack of effect when PM is administered 7 days after bleomycin treatment, which suggests this model is not in keeping with acute exacerbations. However, what is of interest is the role of neutrophils in pathogenesis of lung inflammation and fibrosis and potential therapeutic interventions.

Major concerns:

The bleomycin model used has many limitations. Following bleomycin administration, there is an initial inflammatory response with recruitment of neutrophils and fibrosis starts to develop around day 14. However, my main concern is that the model used in these experiments is not representative of acute exacerbation of IPF. Bleomycin and PM are given simultaneously at day 0. It would be better to use the bleomycin model to establish fibrosis and then challenge with PM (after day 7 or 14 of bleomycin) to be more representative of what occurs clinically in acute exacerbations of IPF. Of note, in the discussion the authors noted that PM given 7 days after bleomycin did not worsen the pulmonary fibrosis. This suggests that the model used is not representative of acute exacerbation and favours PM co-administered with bleomycin augments the fibrotic response. Can the authors amend their current manuscript, including the discussion section. Data in figure 1 shows that combination of bleomycin and PM result in increased fibrosis. There are no data provided to support that bleomycin and PM can cause an acute exacerbation of IPF. For instance, AE-IPF is histologically characterised by haemorrhage and diffuse alveolar damage. Did the lung histology show features of AE-IPF? I suggest the text is amended to reflect that bleomycin and PM result in worsening fibrosis rather than AE-IPF. BAL images provided in figure 2a – difficult to identify neutrophils. The authors confirm previous data that following bleomycin, there is an early increase in BAL neutrophils which resolves by day 7. PM alone increases BAL neutrophils day 2 and this has cleared by day 7. Combination of bleomycin and PM has the greatest effect on increasing BAL neutrophils at day 2 and this has fallen by day 7 with concomitant increase in lymphocytes. Giving sivelestat at early time points after bleomycin +/- PM shows a reduction in the number of neutrophils and collagen. These data suggest that the rapid removal of neutrophils in the initial stages after bleomycin injury, may reduce inflammatory response and the consequent fibrosis. The role of neutrophils in IPF is unclear, but BAL neutrophil count is associated with worse outcome in individuals with IPF. Is the response the same if sivelestat is given later in the injury, for instance at day 2 or day 7 post bleomycin +/- PM when the neutrophil response is declining. Figure 3A – analysis of whole lung – not clear which cells are producing Smad 2/3. Figure 3B – shows neutrophil elastase can activate smad2/3 pathway in primary lung fibroblasts. Is this effect inhibited by sivelestat? KC is a chemokine not a cytokine. Please amend text. Authors have shown that KC is a chemokine and neutrophil chemoattractant. Has BAL from IPF patients been assessed for CXCL8 (the human homolog for KC)? Two sequential doses of reparixin, which inhibits KC binding, given early in the injury (day 1 and 2 post bleomycin +/- PM) reduces BAL neutrophils, neutrophil elastase and collagen. Does this treatment have the same effect if given later in the treatment course. A number of typos throughout the manuscript.

Author Response

(The authors gave the same response as above.)

Round 2

Reviewer 2 Report

The authors have done a good job in addressing the points raised. Most importantly, the data are no longer over-interpreted. In particular, the extrapolation to the clinical situation has been toned down appropriately. 

The significant majority of my comments have been addressed. Inevitably, I still have some residual concerns over a few of the points raised. I do not think it is essential that these be addressed (given the extensive revision already made), but the authors MIGHT wish to consider the following small points which, if addressed, would make their paper a little more balanced.

1. Strictly speaking the model cannot be called IPF...there is nothing idiopathic about a bleomycin bolus. I would recommend changing IPF to PF throughout, when referring to the mouse model.

2. I realise this sounds paradoxical, but while on one hand I think it is very commendable that the authors have gone to the lengths of incorporating lung function, on the other I do think there are a number of problems with the enhanced pause method (Penh). Ideally, the paper would be stronger with one short sentence in the Discussion acknowledging that the method has limitations (perhaps with a reference or two....there is debate in the literature whether it even reflects airway OBSTRUCTION well, let alone RESTRICTION, the characteristic lung function abnormality in pulmonary fibrosis). 

3. The IPF sections (from patients) are easier to see now (in fig S4). I am still not entirely convinced that these show the reader enough to show the characteristic pathological features of UIP. However, there is undoubtedly a lot of fibrosis. The patients are young relative to most with IPF, but perhaps this is why they had biopsies. Ideally, IF the patients had HRCT scans with a UIP pattern, it would be worth inserting this information, to increase readers' confidence that the patients truly had IPF. It is obviously impossible to infer anything about particulate exposure in the human samples, so the human work is still not a perfect fit for the paper, but I understand why it has been included. 

4. I am still not entirely convinced by the composition of the particulate matter. I found the figure hard to follow in the reviewers' response....it seems to imply that quite a lot of the matter is >10um in size, but perhaps I am wrong? However, I accept that the authors used widely available particulate matter, and any other strategy at this stage would be too difficult to incorporate. 

Reviewer 3 Report

The authors have satisfactorily addressed my queries and amended the manuscript as requested. Overall, the manuscript is improved. I have no further queries.